# Concurrence of Marjolin’s Ulcer in the Lower Limb in a Patient with Idiopathic Multicentric Castleman Disease: A Case Report

**DOI:** 10.3390/medicina58010071

**Published:** 2022-01-04

**Authors:** Ping-Ruey Chou, Kun-Bow Tsai, Chao-Wei Chang, Tzu-Yu Lin, Yur-Ren Kuo

**Affiliations:** 1School of Medicine, College of Medicine, Kaohsiung Medical University, Kaohsiung 807, Taiwan; u105025047@gap.kmu.edu.tw; 2Department of Pathology, Kaohsiung Medical University Hospital, Kaohsiung Medical University, Kaohsiung 807, Taiwan; kbtsai@kmu.edu.tw; 3Division of Plastic and Reconstructive Surgery, Department of Surgery, Kaohsiung Medical University Hospital, Kaohsiung Medical University, Kaohsiung 807, Taiwan; jackey_ep@yahoo.com.tw (C.-W.C.); vikuda@gmail.com (T.-Y.L.); 4Faculty of Medicine, College of Medicine, Kaohsiung Medical University, Kaohsiung 807, Taiwan; 5Department of Biological Sciences, National Sun Yat-sen University, Kaohsiung 807, Taiwan; 6SingHealth Duke-NUS Musculoskeletal Sciences Academic Clinical Programme, Singapore 168753, Singapore

**Keywords:** idiopathic multicentric Castleman disease (iMCD), lymph nodal metastasis mimicking, Marjolin’s ulcer in the lower extremity, squamous cell carcinoma, TNM downstage

## Abstract

Idiopathic multicentric Castleman disease (iMCD) is characterized by the benign proliferation of lymphoid cells in multiple regions. However, the co-occurrence of epithelial malignancy and idiopathic multicentric Castleman disease (iMCD) is rarely reported. Herein, we present a case of iMCD mimicking lymph nodal metastasis of Marjolin’s ulcer in the lower extremity. A 53-year-old male presented with an unhealed chronic ulcer on the left lower leg and foot accompanied by an enlarged mass in the left inguinal region. Intralesional biopsy was performed, and pathological examination showed squamous cell carcinoma (SCC). Imaged studies revealed left calcaneus bone invasion, and lymph nodal metastasis was suspected by the cancer TNM staging of T4N2M0 pre-operatively. The patient received below-knee amputation and lymph node dissection; intraoperative histological examination showed no lymphatic nodal malignancy and diagnosed the patient as having iMCD with lymphadenopathy. The patient recovered uneventfully and was referred to a hematologist for further treatment.

## 1. Introduction

Castleman disease (CD), also known as angio-follicular lymph node hyperplasia, is a rare disorder characterized by the benign proliferation of lymphoid cells. Histologically, the CD is subdivided into plasma cell, hyalinized vascular, and intermediate variant types [1]. Clinically, unicentric CD (UCD) and multicentric CD (MCD) are described based on whether enlarged lymph nodes are localized to a single region [2,3]. In recent years, a new classification system [4] has been proposed to further subdivide MCD into HHV8-MCD (human herpes virus-8-associated), POEMS-MCD (polyneuropathy, organomegaly, endocrinopathy, monoclonal, plasma cell disorder, and skin change-associated syndrome), and idiopathic MCD.

Castleman disease, especially idiopathic multicentric Castleman disease (iMCD), reportedly may lead to the development of malignancies [5], the more common types of which include hematological malignancies, such as Kaposi’s sarcoma, follicular dendritic cell sarcoma, and non-Hodgkin and Hodgkin lymphoma [6]. On the other hand, epithelial malignancy is much rarer and seen as an exceptional occurrence [6,7], and its association and co-existence with iMCD have not been well determined. Especially for Marjolin’s ulcer, which reflects malignant degeneration of previously traumatized wounds and is mostly of squamous cell carcinoma (SCC) in the lower limb [8], its concurrence in a patient with iMCD has not been well documented. A unique case of SCC presenting as Marjolin’s ulcer in the lower extremity accompanied by iMCD, which mimics lymph nodal metastasis, is reported here, along with a review of relevant literature regarding diagnosis and clinical management.

## 2. Case Report

A 53-year-old male came to our clinic complaining of a unhealed left leg wound that lasted over many years accompanied by constitutional symptoms of fatigue and loss of appetite. A physical examination revealed excessive granulation tissue with an irregular base and everted wound edges below his left knee covered with necrotic tissue (Figure 1A). An obvious, enlarged palpable mass without tenderness in his left inguinal region was also noted. The remarkable laboratory findings include leukopenia (white cell count 3940/µL), anemia (hemoglobulin 10.6 g/dL), no thrombocytopenia (platelet 281,000/µL), no renal or hepatic dysfunction (urea nitrogen 10.8 mg/dL, creatinine 0.81 mg/dL, aspartate transaminase 37 IU/L, and alanine aminotransferase 29 IU/L); no electrolyte imbalance (sodium 138 mmol/L, potassium 3.9 mmol/L, chlorine 99 mmol/L, and calcium 9.3 mg/dL); immunoglobulin levels (IgG 1440 mg/dL, IgA 961 mg/dL, IgM < 20 mg/dL, β2-microglobulin 204 µg/dL, and normal κ/λ 1.06); negative autoimmune panels (anti-mitochondrial Ab and ASMA); high serum SCC-related antigen (3.4 ng/mL); and no indicative infection of human immunodeficiency virus. Given the chronic and unhealed presentation of the wound, skin biopsy and images for malignancy survey were considered. The Tc-99m MDP whole-body bone scan revealed increased radioactivity in the left distal tibia and especially calcaneus, raising the possibility of bone metastasis (Figure 1B). Additionally, images of contrast-enhanced computed tomography (CT) displayed multiple stations of enlarged lymph nodes along the left iliac chain and obturator space (Figure 1C), the left inguinal region, and the left popliteal fossa (Figure 1D), indicating possible lymph nodal metastasis. SCC was then confirmed by microscopic examination showing invasive nests of large, keratinizing epidermoid cells (Figure 2A), and its clinical presentation favored Marjolin’s ulcer. The stage of SCC was preliminarily determined as T4N2M0 [9]. Under the impression of Marjolin’s ulcer with bone metastasis and lymphadenopathy, he underwent left below-knee amputation and left inguinal lymph node dissection. The pathology report showed the characteristics of plasma-cell type CD with hyperplastic germinal centers surrounded by an “onion-skin”-like arrangement of lymphocytes (Figure 2B) and prominent inter-follicular plasma cells proliferation (Figure 2C). HHV8-MCD was excluded by immunohistochemistry (IHC) detection (Figure 2D). There was no kappa and lambda light chain restriction, ruling out plasma cell myeloma (Figure 2E,F). Accordingly, the cancer stage was revised to T4N0M0 [9], reaching a final diagnosis of SCC in the left lower leg and foot without lymph nodal metastasis, and a diagnosis of iMCD was also made based on the diagnostic criteria [6]. The patient recovered uneventfully after the surgery and received further management of iMCD under our Hematology outpatient department follow-up for about one and a half years.

## 3. Discussion

Castleman disease (CD) was first reported by Castleman et al. as a rare disorder featuring benign lymphoproliferation and the constellation of enlarged lymph nodes in one region, most commonly the mediastinum [10,11]. Gaba et al. later recognized that the presentation of CD might be unicentric (UCD) or multicentric (MCD) [12]. In contrast to UCD, MCD is more prevalent in males and in older individuals [13]. Opportunistic HHV8 infection arising from an HIV-1-immunocompromised state is one of the major risk factors for MCD patients and frequently causes malignancy, such as Kaposi’s sarcoma [14], of which the histopathological type is plasma cell and plasmablastic [15].

Another subset of CD, termed iMCD, has clinicopathological features that overlap with those of HHV8-MCD but have no known etiology [5]. iMCD was difficult to diagnose in the past due to its wide range of poorly defined symptoms and laboratory abnormalities. In 2020, Dispenzieri et al. [6] proposed a set of diagnostic criteria for iMCD consisting of two major defining characters: (1) distinct histopathological characteristics such as hyperplastic germinal centers and prominent inter-follicular plasma cells proliferation, and (2) enlarged lymph nodes in at least two anatomical regions and in at least two minor ones (constitutional symptoms and at least one laboratory abnormality). iMCD can be further subdivided into iMCD-TAFRO (thrombocytopenia, ascites, reticulin fibrosis, renal dysfunction, and organomegaly) and iMCD-NOS (not otherwise specified) [4]. iMCD-TAFRO patients tend to have more aggressive clinical manifestations, whereas those who do not meet the criteria of iMCD-TAFRO [6] and who have less aggressive presentation are classified into iMCD-NOS. In the present case, the histopathological results, images of lymph nodes, and laboratory data with constitutional symptoms met the diagnostic criteria of iMCD but not iMCD-TAFRO. HHV8-MCD, HIV infection, and multiple myeloma were also ruled out. Thus, the diagnosis of iMCD was accordingly confirmed, and the subtype was more compatible with iMCD-NOS.

Malignant diseases are not commonly associated with CD and the risk of malignancy in patients with iMCD is still unclear. However, iMCD has a threefold increase in the frequency of malignancy compared with age-matched controls [5]. The exact pathogenesis of co-occurrence of malignancy and iMCD is still not fully determined, but several hypotheses exist [5]. First, iMCD might be malignancy-driven. Malignant cells produce interleukin-6 (IL-6) and other proinflammatory cytokines that can potentially drive the symptoms of iMCD. Second, iMCD might be a presentation of the progressing premalignant state. Last, genetic mutations or other unidentified infections might enhance the susceptibility to iMCD and malignancies. Among the malignancies, those of epithelial origin are exceptionally rare and previously mentioned only in a few cases reports, all presenting lymph nodal metastasis-like iMCD, including pulmonary carcinoma [16,17] and hypopharyngeal carcinoma [18]. The present case is a rare report of the left lower leg and foot SCC presenting as Marjolin’s ulcer with iMCD mimicking lymph nodal metastasis. Although there was a high risk of lymph nodal metastasis of Marjolin’s ulcer [19], the lymph node histological report showed that the iMCD had a benign nature, and the cancer staging was from T4N2M0 to T4N0M0, which guided different subsequent managements. With the diagnosis of iMCD, systemic therapeutic monoclonal antibodies are recommended by the consensus guidelines proposed by Rhee et al. [20], and there is no need for adjuvant radiotherapy or chemotherapy after the surgery was suggested under the initial impression of SCC lymph nodal metastasis.

In summary, we reported an extremely rare case of iMCD with SCC presenting as Marjolin’s ulcer in the lower extremity. Lymph nodal metastasis was initially suspected but down-staged by the histological iMCD diagnosis, which guided different subsequent managements. Thus, we suggest that benign lymphoid diseases, such as iMCD, should be taken into consideration in patients with suspected lymph nodal metastasis from epithelial malignancy.

## Figures and Tables

**Figure 1 medicina-58-00071-f001:**
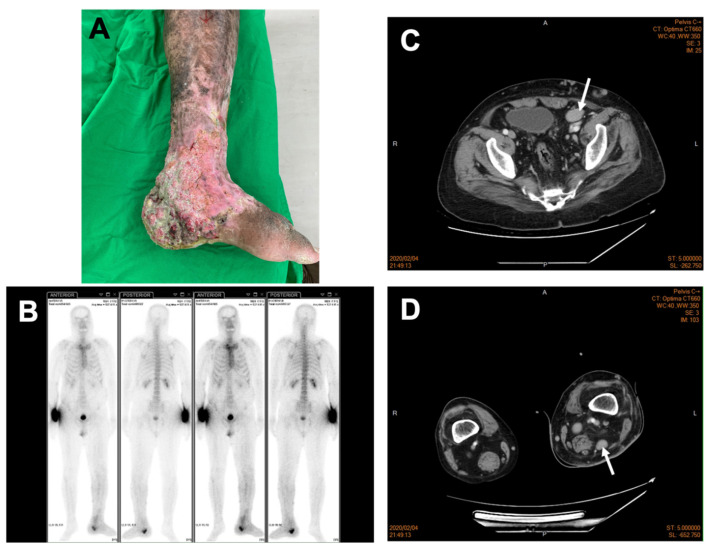
Preoperative image evaluation of the left lower leg and foot SCC. (**A**) SCC presented as pigmented skin and chronic ulcers below the left knee, especially at ankle and heel; (**B**) Tc-99m MDP whole-body bone scan revealed increased radioactivity in the left calcaneus, extraosseously over the left lower leg and foot (due to soft tissue swelling), calvarium, thoracolumbar spine (due to certain degenerative change), and focal radiotracer accumulation in the right wrist, which may indicate injection sites; and contrast-enhanced computed tomography showed enlarged and enhanced lymph nodes along (**C**) the left external iliac vessels (greatest diameter: 3.16 cm, white arrow) and (**D**) the left popliteal fossa (greatest diameter: 2.13 cm, white arrow).

**Figure 2 medicina-58-00071-f002:**
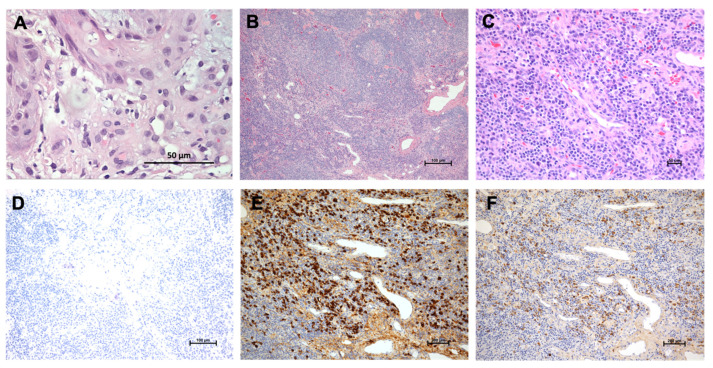
Histopathological examination. (**A**) SCC with invasive nests of large, keratinizing epidermoid cells is shown (HE × 400). The malignancy nature was identified by peer slide review; (**B**) the “onion-skin”-like layer of lymphocytes surrounding the hyperplastic germinal center (HE × 40); (**C**) prominent sheets of plasma cell proliferation in the interfollicular zone (HE × 200); (**D**) no HHV8-positive cells were detected to exclude HHV8-MCD (IHC × 100); and no kappa (**E**) (IHC × 100) and lambda (**F**) (IHC × 100) restriction shown to exclude plasma myeloma malignancy.

## Data Availability

The original contributions presented in the study are included in the article. Further inquiries can be directed to the corresponding author.

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
