# Peer review of "Concurrence of Marjolin’s Ulcer in the Lower Limb in a Patient with Idiopathic Multicentric Castleman Disease: A Case Report"

_medicina, 2022, doi:10.3390/medicina58010071_

Round 1

Reviewer 1 Report

Originality:  This is a very short paper, providing a minimum of information about a case study, and constitutes a very minor contribution to the literature.  The introduction section did not provide a clear rationale for carrying out the study (for example, why is your research question important? What gap in the literature is the study addressing?). I suggest to describe in this section only with the information related with the state of art related with Marjolin’s ulcer in the lower limb related with idiopathic multicentric Castleman disease

Methodologically Sound:  As a case study report it is rather hard to go wrong methodologically, and the paper conforms to the standard.

Follows Appropriate Ethical Guidelines: Whilst there is no obvious declaration of ethical approval. Please include the date and code register number of ethics committee, it would appear to be a report of actions taken as part of normal clinical practice (as a case study report), and thus is acceptable. 

Has results which are clearly presented and support the conclusions: Again, it conforms to the usual format for the presentation of a case study, although the content is very sparse.  It is, however, appropriate enough, and does report a rare case likely to be of interest to a healthcare audience.  

Overall Scientific Quality:  As a minor case study report it lacks scientific depth, but effectively is intended only to report the occurrence of a typical case and to highlight the importance of correct diagnosis, and on these grounds merits attention. 

Presentation, Organization, Clarity:  I think you have some good information. But it is poorly presented. 

Correctly References Previous Relevant Work:  It appears to reference prior work succinctly and accurately. 

Importance/Interest: Although marked by its brevity, the content is of interest, particularly to clinicians such as  plastic surgeons and in the reconstructive Surgery  who examine orthopaedic problems  who may need to be aware of the variant forms of this illness.  

OVERALL ASSESSMENT:  Some important information; needs major change;  priority medium

Author Response

Thank you very much for your letter regarding our manuscript (medicina-1506978-R1) entitled “Concurrence of Marjolin’s ulcer in the lower limb in a patient with idiopathic multicentric Castleman disease: A case report”. We have carefully followed the reviewer’s suggestions to revise this article accordingly. The revised parts have been highlighted with track changes in the manuscript. We now address the changes we have made, point-by-point, to the reviewers’ comments as follows.

Reviewer 1:

Q1: Originality:  This is a very short paper, providing a minimum of information about a case study, and constitutes a very minor contribution to the literature.  The introduction section did not provide a clear rationale for carrying out the study (for example, why is your research question important? What gap in the literature is the study addressing?). I suggest to describe in this section only with the information related with the state of art related with Marjolin’s ulcer in the lower limb related with idiopathic multicentric Castleman disease.

Response: As your concern, we agree this case report might not have a big contribution to the literature. However, idiopathic multiple Castleman disease (iMCD) with epithelial malignancy such as Marjolin’s ulcer is extremely rare reported. Also concurred lympho-adenopathy is hard to make a different diagnosis of lymph node metastasis or only lymphoid hyperplasia related with iMCD for cancer staging and therapy.  We have described the information of Marjolin’s ulcer in the lower limb related with idiopathic multicentric Castleman disease in line 6-9 in the 2nd paragraph of Introduction section.

Q2: Methodologically Sound:  As a case study report it is rather hard to go wrong methodologically, and the paper conforms to the standard.

Response:    Thank you for your comment on “Methodologically Sound”. We have revised the manuscript to the paper conform.

Q3: Follows Appropriate Ethical Guidelines: Whilst there is no obvious declaration of ethical approval. Please include the date and code register number of ethics committee, it would appear to be a report of actions taken as part of normal clinical practice (as a case study report), and thus is acceptable. 

Response:  As your concern, we have added the date and code register number of ethics committee in line 2 of Ethics statement section.

Q4: Has results which are clearly presented and support the conclusions: Again, it conforms to the usual format for the presentation of a case study, although the content is very sparse.  It is, however, appropriate enough, and does report a rare case likely to be of interest to a healthcare audience.  

Response:   Thank you for your positive advice. We have revised the manuscript and conform the usual format as case study

Q5: Overall Scientific Quality:  As a minor case study report it lacks scientific depth, but effectively is intended only to report the occurrence of a typical case and to highlight the importance of correct diagnosis, and on these grounds merits attention. 

Response:    As your comments, we agree this report might lack scientific depth but worthy to highlight the importance of correct diagnosis.

Q6: Correctly References Previous Relevant Work:  It appears to reference prior work succinctly and accurately. 

Response:  Thank you for your suggestion, we have revised that in the manuscript.

Q7: Importance/Interest: Although marked by its brevity, the content is of interest, particularly to clinicians such as plastic surgeons and in the reconstructive Surgery who examine orthopaedic problems who may need to be aware of the variant forms of this illness.  

Response:  Thank you for your suggestion, indeed this is the key points to be aware of the variant forms of this illness.

Reviewer 2

Q1: It is still unclear if patients with iMCD have higher risks of malignancy - the reference you cited (ref 5) was published before 2017 when the International Diagnostic Criteria for iMCD, which mandates us to exclude hematologic malignancies, was proposed. Thus, the sentence needs to be revised, and it would be better to state the risk of malignancy in patients with iMCD is unclear. 

Response:    As your suggestion, we have revised the sentence in line 1-2 in the 3rd paragraph of Discussion section.

Q2: iMCD needs to be further characterized into either iMCD-NOS or iMCD-TAFRO based on the recently proposed international definition. It seems like the case could be iMCD-NOS. Still, the authors need to describe detailed laboratory findings including CBC, BMP, LFTs, IL-6, VEGF, immunoglobulin levels, and autoimmune panels such as ANA, dsDNA, SS-A, SS-B, and ANCA.

https://onlinelibrary.wiley.com/doi/abs/10.1002/ajh.26292

Response:  Thank you for your comment. We have revised and included all existing and available laboratory data aiding iMCD diagnosis in line 6-14 of Case report section.

Q3: Please specify the follow-up period. 

Response: As your suggestion, the patient follow-up around one and half year.  We have described in line 34-35 of Case report section.

To close, we would again like to thank you for the suggestions to improve our manuscript. We hope that the revised manuscript is suitable for publication.

Sincerely yours,

Dr. Yur-Ren Kuo M.D., Ph.D., FACS,

Division of Plastic & Reconstructive Surgery, Department of Surgery,

Kaohsiung Medical University Hospital, Taiwan

100 Tzyou 1st Rd., Kaohsiung 80756, Taiwan.

Phone: +886-7-3121101, ext. 7675, Fax: +886-7-7311482

Reviewer 2 Report

The case report by Chou et al. describes an interesting case of SCC as Marjolin's ulcer with coincidental iMCD. While the case illustrates the critical point regarding cancer staging in systemic inflammatory diseases, several issues need to be addressed before publications. 

1. It is still unclear if patients with iMCD have higher risks of malignancy - the reference you cited (ref 5) was published before 2017 when the International Diagnostic Criteria for iMCD, which mandates us to exclude hematologic malignancies, was proposed. Thus, the sentence needs to be revised, and it would be better to state the risk of malignancy in patients with iMCD is unclear. 

2. iMCD needs to be further characterized into either iMCD-NOS or iMCD-TAFRO based on the recently proposed international definition. It seems like the case could be iMCD-NOS. Still, the authors need to describe detailed laboratory findings including CBC, BMP, LFTs, IL-6, VEGF, immunoglobulin levels, and autoimmune panels such as ANA, dsDNA, SS-A, SS-B, and ANCA.
https://onlinelibrary.wiley.com/doi/abs/10.1002/ajh.26292

3. Please specify the follow-up period. 

Author Response

(The authors gave the same response as above.)

Round 2

Reviewer 1 Report

The manuscript reads much better and I am happy to recommend it for publication.
Good luck to you all!

Author Response

Thank you very much for your letter regarding our manuscript (medicina-1506978-R2) entitled “Concurrence of Marjolin’s ulcer in the lower limb in a patient with idiopathic multicentric Castleman disease: A case report”. We have carefully followed the reviewer’s suggestions to revise this article accordingly. The revised parts have been highlighted with track changes in the manuscript. We now address the changes we have made, point-by-point, to the reviewers’ comments as follows.

Reviewer 1:

Q: The manuscript reads much better and I am happy to recommend it for publication. Good luck to you all!

Response: Thank you for your positive comments and recommendation.

Reviewer 2

Q1: Thank you for your revision. As I suggested in the previous round, please describe in detail if the case was more compatible as iMCD-NOS or iMCD-TAFRO, as well as adding an explanation to these subtypes in your introduction or discussion. 

Response: Thank you for your suggestion, we have further described in detail about the reason why our case was more compatible with iMCD-NOS in the subtype. The revision can be traced in line 8-17 in the 2nd paragraph of Discussion section.

To close, we would again like to thank you for the suggestions to improve our manuscript. We hope that the revised manuscript is suitable for publication.

Sincerely yours,

Dr. Yur-Ren Kuo M.D., Ph.D., FACS,

Division of Plastic & Reconstructive Surgery, Department of Surgery,

Kaohsiung Medical University Hospital, Taiwan

100 Tzyou 1st Rd., Kaohsiung 80756, Taiwan.

Phone: +886-7-3121101, ext. 7675, Fax: +886-7-7311482

Reviewer 2 Report

Thank you for your revision. As I suggested in the previous round, please describe in detail if the case was more compatible as iMCD-NOS or iMCD-TAFRO, as well as adding an explanation to these subtypes in your introduction or discussion. 

Author Response

(The authors gave the same response as above.)
